# Development of a theoretical framework of factors affecting patient safety incident reporting: a theoretical review of the literature

Stephanie Archer,[1] Louise Hull,[1,2] Tayana Soukup,[1] Erik Mayer,[1] Thanos Athanasiou,[1] Nick Sevdalis,[1,2] Ara Darzi[1]

[1]NIHR Imperial Patient Safety Translational Research Centre, Imperial College London, London, UK
[2]Centre for Implementation Science, King's College London, London, UK

**Correspondence to**
Dr Stephanie Archer; stephanie.archer@imperial.ac.uk

## ABSTRACT

**Objectives** The development and implementation of incident reporting systems within healthcare continues to be a fundamental strategy to reduce preventable patient harm and improve the quality and safety of healthcare. We sought to identify factors contributing to patient safety incident reporting.

**Design** To facilitate improvements in incident reporting, a theoretical framework, encompassing factors that act as barriers and enablers ofreporting, was developed. Embase, Ovid MEDLINE(R) and PsycINFO were searched to identify relevant articles published between January 1980 and May 2014. A comprehensive search strategy including MeSH terms and keywords was developed to identify relevant articles. Data were extracted by three independent researchers; to ensure the accuracy of data extraction, all studies eligible for inclusion were rescreened by two reviewers.

**Results** The literature search identified 3049 potentially eligible articles; of these, 110 articles, including >29 726 participants, met the inclusion criteria. In total, 748 barriers were identified (frequency count) across the 110 articles. In comparison, 372 facilitators to incident reporting and 118 negative cases were identified. The top two barriers cited were fear of adverse consequences (161, representing 21.52% of barriers) and process and systems of reporting (110, representing 14.71% of barriers). In comparison, the top two facilitators were organisational (97, representing 26.08% of facilitators) and process and systems of reporting (75, representing 20.16% of facilitators).

**Conclusion** A wide range of factors contributing to engagement in incident reporting exist. Efforts that address the current tendency to under-report must consider the full range of factors in order to develop interventions as well as a strategic policy approach for improvement.

## BACKGROUND

The development and implementation of incident reporting systems within healthcare continues to be a fundamental strategy to reduce preventable patient harm and improve the quality and safety of healthcare on a local,

### Strengths and limitations of this study

► The synthesis included quantitative, qualitative and mixed methods research and was not restricted to specific incident reporting systems.
► Only articles published in English were included.
► The last systematic search for literature was conducted on 29 May 2014, meaning that literature published since this date will not have been included.
► Studies detailing interventions to improve incident reporting and studies detailing variations in engagement in incident reporting were not included.
► Large heterogeneity across studies in terms of outcome measures and methodologies meant conduction of meta-analysis was precluded.

regional and national basis.[1 2] Although coverage and sophistication vary widely, incident reporting systems have now been in place for more than a decade in a number of countries.[3]

A key factor that compromises the ability of incident reporting systems to improve patient safety is under-reporting. In the USA, it is estimated that 50%–96% of incidents are not reported.[2 4 5] Failure to report patient safety incidents significantly hinders the underlying goals of incident reporting systems; low levels of reporting make it difficult at best to identify and prioritise patient safety risks and hamper learning from such incidents and ultimately improvements in patient safety. While debate continues to exist regarding whether all patient safety incidents should be reported,[6 7] it is extremely important to understand the factors that act as barriers and facilitators to incident reporting so that 'sufficient' levels of reporting exist to facilitate learning and improvement.

A number of studies exploring barriers and facilitators to incident reporting have been conducted.[8–11] In addition, a number

of literature reviews to identify barriers and facilitators to incident reporting have been published.[12–14] Although previous work has made a valuable contribution to our understanding of factors affecting incident reporting, previous work has been limited in scope (eg, focusing on the psychological factors affecting incident reporting;[14] focusing on perceived barriers influencing incident reporting by nurses;[13] factors affecting reporting of incidents related to medical devices and other healthcare technologies).[12] As such, to date, there has been no definitive synthesis and evaluation of the factors that prevent or promote reporting.

The primary aim of this theoretical review was to systematically identify factors affecting patient safety incident reporting. The secondary aims were, first, to develop a theoretical framework of factors acting as barriers and facilitators to incident reporting to guide implementation of interventions to increase engagement, and, second, to determine the prevalence of factors to guide the development of interventions and policies to improve incident reporting.

## METHODS
### Theoretical review

A theoretical review was conducted as the overarching goal of the review was to build explanation of factors affecting incident reporting. In line with a theoretical review, both quantitative and qualitative data were eligible for inclusion and interpretive methods were used to synthesise findings.

### Study searches and selection

A systematic search strategy was developed and an electronic search was carried out in three databases: Embase, Ovid MEDLINE(R) and PsycINFO. The last search was conducted on 29 May 2014; while the last search was conducted 3 years ago, this reflects the sheer volume of articles that were included in this review. Search terms included those related to patient safety incidents, incident reporting systems, and barriers and facilitators to engagement in reporting (see table 1 for full search terms). Time and language of publications was restricted from 1980 and to English language.

### Eligibility criteria
#### Inclusion criteria
1. Studies reporting factors influencing the likelihood of incident report engagement in any healthcare setting (eg, primary and secondary healthcare) and employing any study design (eg, qualitative, quantitative, mixed methods).

#### Exclusion criteria
1. Studies reporting aspects of incident reporting systems and/or incident reporting perceived positively and/or negatively by healthcare professionals (HCPs)

**Table 1** Search strategy

| | |
|---|---|
| Category A | Patient safety incident: near adj miss* (MeSH heading), adverse adj event*, never adj event* (MeSH entry term), medical adj mistake* (MeSH entry term), error*, mistake* (MeSH entry term), negligen* (MeSH entry term), malpractice* (MeSH entry term), failure*, injur* (MeSH entry term), critical adj incident* (MeSH entry term), sentinel adj event*, incident*, harm*, accident* (MeSH heading), medical adj error* (MeSH heading), patient adj safety (MeSH heading) |
| Category B | Incident reporting system: risk adj management (MeSH heading), incident adj reporting adj system*, error adj report*, critical adj incident adj technique (MeSH entry term), safety adj report* (MeSH entry term), incident adj report* (MeSH entry term), reporting adj system, NRLS, national adj reporting adj2 learning adj system. |
| Category C | Barrier/facilitator: communication adj barrier* (MeSH heading), feedback (MeSH heading), safety adj culture (MeSH entry term), reporting adj culture, attitude (MeSH heading)*, preventive adj measure* (MeSH entry term), mandatory, voluntary, under-reporting, willingness, blame, obstacle*, incident adj type, level adj of adj harm, fear* (MeSH heading), responsibi*, workload (MeSH heading), anonym*, confidential* (MeSH heading), trust* (MeSH heading), anonym*, confidential* (MeSH heading), facilit*, barrier*, enabl*, legal, law (MeSH entry term). |

without data relating perceptions to incident reporting engagement.

2. Studies reporting data relating to disclosure of patient safety incidents to patients or their families (a systematic review of the literature on patient/family disclosure has previously been published).[15]

3. Studies reporting data relating to the effectiveness of interventions to improve incident reporting (a systematic review of the literature on the effectiveness of interventions to increase clinical incident reporting in healthcare has previously been published).[13]

4. Studies reporting statistical models where the impact of individual barriers and facilitators to engagement in incident reporting was unable to be determined.

The eligibility criteria were developed to maintain a focus on factors having a direct impact on incident reporting engagement rather than simply identifying and listing factors of incident reporting which were perceived positively or negatively by HCPs. Identifying elements of incident reporting perceived positively or negatively by HCPs does not equate to identifying factors that have an impact on reporting behaviour. In such studies, it is not possible to determine the impact on reporting behaviour—the primary focus of this review.

### Data extraction

After the removal of duplicates, two authors (SA and LH) independently reviewed all articles on the basis of the titles and abstracts. Three authors (SA, LH and TS) reviewed the articles at full-text stage. Data were extracted using an extraction template. The following data were extracted: first author's name, year of publication, country, study design, study population, sample size and factors that decrease (barriers), increase (facilitators) or were neither a barrier nor facilitator to engagement in incident reporting (negative cases). To ensure the accuracy of data extraction, all studies eligible for inclusion were rescreened by two reviewers (SA and LH).

### Quality assessment

Many assessment tools and checklists have been developed to appraise the quality and susceptibility to bias of studies (eg, the Cochrane Collaboration's tool for assessing risk of bias in randomised trials;[16] AMSTAR tool to assess the methodological quality of systematic reviews;[17] tools to assess the quality of qualitative research studies).[18] The decision not to assess the quality of studies was made for a number of reasons. First, the large heterogeneity of study designs would have made comparisons between study designs difficult at best. Second, quality appraisal is not considered necessary for theoretical reviews.[19] Third, it has been argued that it is important, but difficult, to distinguish between 'quality of reporting' and the 'quality of a study'.[20] As such, articles were not excluded from the current review based on 'quality' nor was weight assigned to studies based on quality.

### Data analysis and initial theoretical framework development

A grounded theory approach was used to guide the development of the theoretical framework. Grounded theory is associated with the discovery of theory from data systematically obtained from social research.[21] It has been identified as a method where thorough and theoretically relevant analysis of a topic can be reached, specifically within literature reviews.[22] In light of this, a three-stage approach was undertaken to develop a theory of factors contributing to engagement in patient safety incident reporting. The first stage, *coding*, includes identifying parts of the data that relate the phenomena in question (in this case, incident reporting). During this stage, known as *open coding* in the grounded theory literature, three authors (SA, LH and TS) read and re-read each paper and identified sections of the paper that were relevant to the research question. Initial concepts developed from these were noted down at this stage; in some cases, these were consistent with pre-existing literature (eg, in the case of a standardised scale), but in others allowed for unseen insights to develop across the data corpus (eg, in qualitative studies). In the second stage, *conceptualising*, or *axial coding*, focused on grouping together the initial codes where there were relationships to form higher-order categories. These were given names. Stage 3, *categorising*, or *selective coding* focused on linking together similar higher-order categories that contained similar concepts which could underpin the reasoning behind the way that the phenomena (in this case, incident reporting) could be explained. Figure 1 displays an example of how these stages were applied.

Engagement in these three stages allowed constant comparison between the articles in the data set to be performed until a theoretical framework was confirmed.

The final theoretical framework was reviewed by another member of the research team (NS) and feedback regarding the category descriptors was incorporated. The final theoretical framework of factors contributing to patient safety incident reporting engagement is displayed in table 2.

The theoretical framework developed was used to organise the identification of factors found to affect incident reporting and to quantify their prevalence. This approach is consistent with existing frameworks in the patient safety literature. For example, Lawton *et al* employed a similar approach to quantify the prevalence of factors contributing to patient safety incidents in hospital settings.[23]

### Patient and public involvement

No patients were involved in setting the research question or the outcome measures, nor were they involved in the design and implementation of the study. We do not anticipate patients and the public being involved in the dissemination of the work.

### Findings

The search identified 5335 records. After duplicates and limits were applied (English language, date restrictions

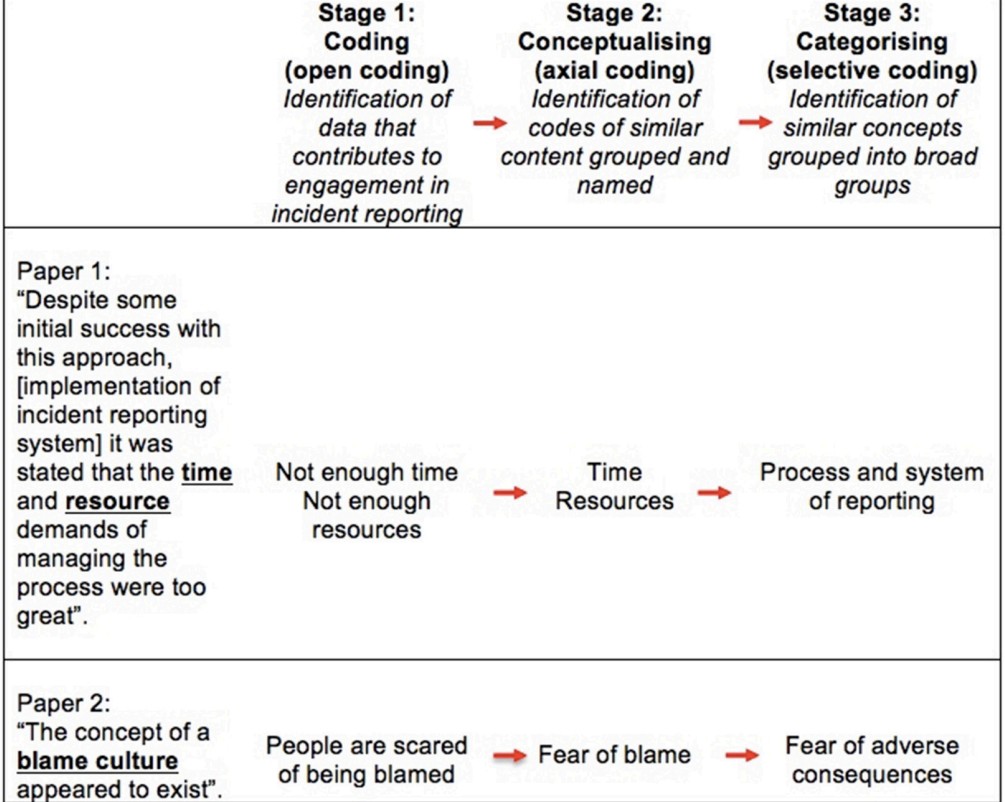

**Figure 1** Example of data coding, conceptualisation and categorisation for theory development.

1980–May 2014), 3049 records were considered for inclusion. Of these 3049 records, 2700 were excluded based on title and abstract screening. A total of 349 articles were considered potentially relevant and were assessed at full-text by two researchers (kappa 0.70, p<0.001). Of 349 publications, 33 were not obtainable (requested through the British Library), leaving 314 articles assessed at full-text stage. From these, 80 articles met inclusion criteria.

The reference lists of all included articles were screened for potentially relevant publications, resulting in a further 30 articles that met the inclusion criteria. A total of 110 articles, including >29 726 participants, were included in the final review (figure 2). The total number of participants per study ranged from 8 to 2185 (mean=285.83; median: 134.00). Six studies did not report sample size, thus the sample size calculations represented above are based on 104 articles.[24–29] See online supplementary table 1 for full data extraction.

### Study characteristics
#### Empirical study types and design
In total, 110 articles were included; these consisted of 76 quantitative studies (including 72 questionnaire-based studies, 1 secondary analysis of data study, 1 case control study, 1 descriptive study and 1 cohort study), 21 qualitative studies (including 11 interview-based studies and 10 focus group studies) and 13 mixed-methods studies (1 semistructured interview and documentary analysis-based study; 1 semistructured interview and retrospective review

of error reports-based study; 2 semistructured interview and questionnaire-based studies; 3 focus group and questionnaire-based studies; 1 semistructured and structured interview-based study; 1 interview, focus group and analysis of event reports-based study; 1 focus group and semistructured interview-based study; 1 retrospective analysis of routinely collected data and questionnaire-based study; 2 focus groups, interview and questionnaire-based studies).

#### Countries
The review encompassed research spanning 4 continents and >20 countries. The four countries contributing the most studies were the USA (n=33), the UK (n=24), Australia (n=8) and Canada (n=8), table 3.

#### Year of publication
A steady increase in articles was evident over decades: 1980s (n=1),[30] 1990s (n=12),[24 31–41] 2000s (n=58),[8–11 28 29 42–93] and 2010–May 2014 (n=39).[25–27 94–129] This increase is likely to reflect the growing integration of incident reporting systems in healthcare systems worldwide and the increasing realisation that HCPs' engagement in incident reporting is far from ideal.

The frequency of barriers and facilitators to incident reporting across the 110 articles was calculated and rank ordered across the data (figure 3). Where contributing factors were found not to be barriers or facilitators to incident reporting (eg, if fear was found not to be a

**Table 2** Theoretical framework of factors determining engagement in patient safety incident reporting

| Category | Descriptions and examples |
|---|---|
| Organisational | Organisational values, beliefs and policies around incident reporting. This also encompasses any organisational factor which may act as a barrier or facilitator to reporting behaviour, such as structure (eg, size of hospital) and organisational culture. |
| Work environment | Features of the work environment that act as barriers or facilitators to engagement in incident reporting. Examples of such factors include level of activity, staffing levels and visual prompts. |
| Process and systems of reporting | Any characteristics or features of the reporting system/process which enables or hinders incident reporting. This includes the complexity of the reporting system, the level of information required and the mode of incident reporting (eg, paper based or electronic). |
| Team factors | Any factor related to the functioning of different professionals within a group which influences incident reporting behaviour. For example, support and encouragement by team members to report incidents, and levels of teamwork and communication. |
| Knowledge and skills | The acquisition and development of knowledge and skills that enables incident reporting. This includes participation in specific (eg, form completion) and general (eg, identifying which incidents warrant reporting) training/educational activities. |
| Individual HCP characteristics | Characteristics of the HCP that may contribute in some way to engagement in incident reporting. Examples of such factors include seniority, personality and attitudes. |
| Professional ethics | The accepted standards of personal and professional behaviour, values and guiding principles that promote incident reporting. For example, the adoption of sound and consistent ethical practices, such as duty of care. |
| Fear of adverse consequences | Any unpleasant emotion (eg, guilt) or outcome (eg, litigation) associated with individual HCPs' incident reporting behaviour. A reduction in the likelihood of experiencing fear (eg, the existence of a non-punitive policy) results in increased incident reporting participation. |
| Incident characteristics | Characteristics of the patient safety incident which may make HCPs more or less likely to report. These include frequency of error, level of harm and the cause of error. |

HCP, healthcare professional.

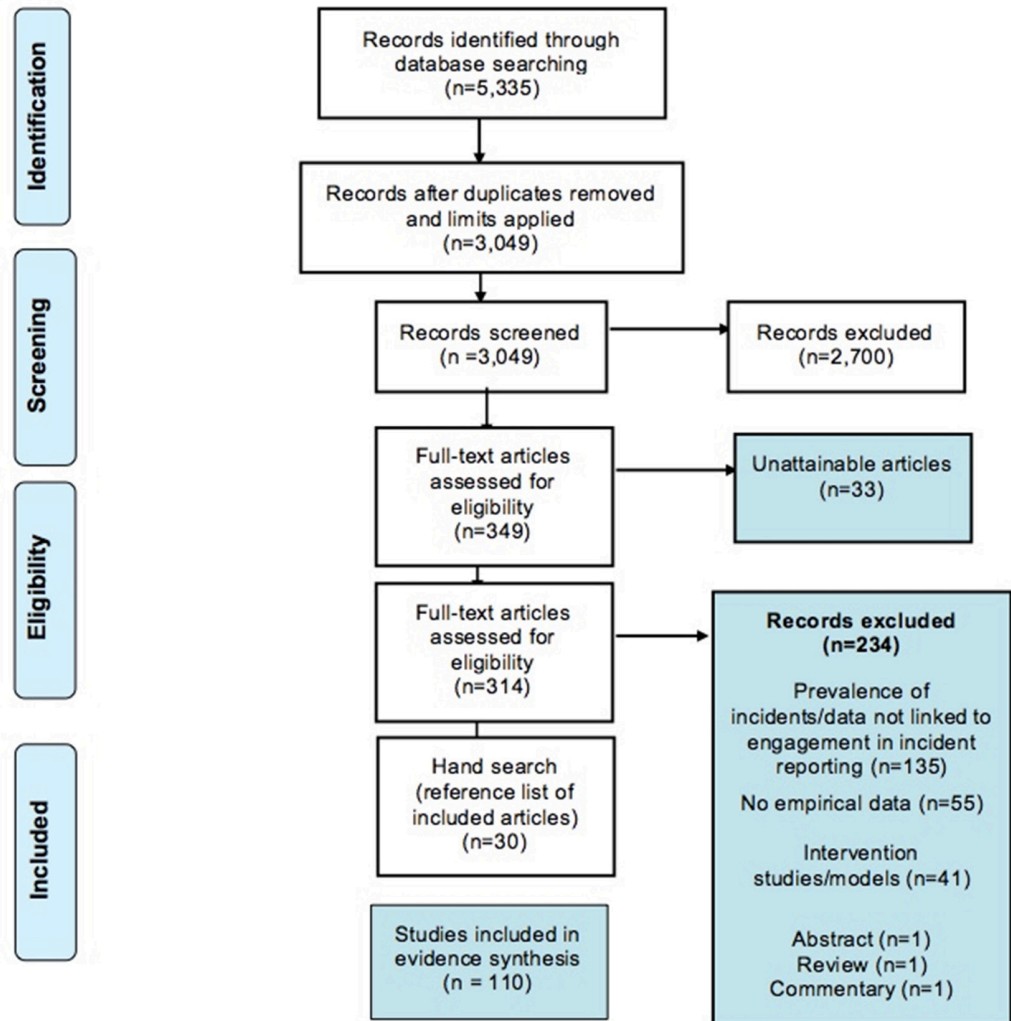

**Figure 2** Flow diagram of the theoretical literature review process.

significant predictor of decreased or increased incident reporting), these were counted as negative cases. These negative cases were included to provide a more complete view of the data, and to prevent reporting bias.

When the same barrier, facilitator or negative case (eg, fear of adverse consequences) was mentioned more than once within an article, this was reflected in the frequency data presented. In total, 748 barriers to incident reporting were identified (frequency count) compared with 372 facilitators. A total of 118 negative cases were identified. The top two barriers cited were fear of adverse consequences (161, representing 21.52% of barriers) and process and systems of reporting (110, representing 14.71% of barriers). In comparison, the top two facilitators were organisational (97, representing 26.08% of facilitators) and process and systems of reporting (75, representing 20.16% of facilitators). These results illustrate that the factors identified in this review of the literature can act as both a barrier and a facilitator to incident reporting systems depending on context; for example, *process and systems of reporting* was found to be the second most frequently cited barrier, as well as the second most

frequently cited facilitator to incident reporting engagement. While this may initially appear contradictory, when considering the complexity/simplicity of reporting it was found that highly complex incident reporting processes and systems were a barrier to incident reporting, whereas simple processes and systems were found to be a facilitator.

### Frequency of barriers to patient safety incident reporting

Barriers to incident reporting were mentioned 748 times across the 110 articles (see online supplementary table 2). The three most frequently mentioned barriers to incident reporting included *fear of adverse consequences* (161/748), *process and systems of reporting* (110/748) and *incident characteristics* (92/748).

### Fear of adverse consequences

Fear of adverse consequences, as a barrier, was mentioned 161 times and included a general fear of adverse consequences associated with incident reporting (51/161),[8 10 11 27 31 33 39 41 42 44 45 47 48 51–53 59–61 63 64 69 71 72 76 79 87 94 98 102 108 109 114 116 118 120 122 126] fear of litigation (30/161),[8–11 24 27 30 32 35 37–40 44 47 56 67 70 77 80 81 84 86 92 98 108 109 116 119 129] and

**Table 3** Frequency of articles by country

| Country | Count (%) |
| --- | --- |
| USA[9 11 28 30–33 42–64 94–96] | 33 (30.00) |
| UK[10 29 34–38 65–72 97–105] | 24 (21.82) |
| Australia[8 27 39 73 74 106–108] | 8 (7.27) |
| Canada[75–78 109–112] | 8 (7.27) |
| Taiwan[79 113–115] | 4 (3.64) |
| Netherlands[40 80 116 117] | 4 (3.64) |
| Saudi Arabia[81 118–120] | 4 (3.64) |
| International[24 26 121 122] | 4 (3.64) |
| Israel[82 83 123] | 3 (2.73) |
| Iran[84 124] | 2 (1.82) |
| Japan[25 125] | 2 (1.82) |
| New Zealand[85 86] | 2 (1.82) |
| Sweden[87 88] | 2 (1.82) |
| Italy[41 126] | 2 (1.82) |
| Denmark[127] | 1 (0.91) |
| Norway[128] | 1 (0.91) |
| Pakistan[129] | 1 (0.91) |
| Portugal[89] | 1 (0.91) |
| Jordan[90] | 1 (0.91) |
| China[91] | 1 (0.91) |
| Germany[92] | 1 (0.91) |
| Spain[93] | 1 (0.91) |

the fear of blame (24/161).[8 10 35 44 47 52–54 63 64 71–73 76 79 97 98 102 103 108 111 120] Additionally, the fear of judgement (22/161),[10 24 34 37 47 52 59 64 72 76 79 81 85 90 109 118 122] the fear of the negative impact that incident reporting could have on relationships with other HCPs, patients and the public (12/161),[10 11 33 53 54 56 64 76 85 94 118 126] and the fear of a detrimental impact that reporting an incident could have on HCPs' career (10/161),[10 11 27 63 64 72 76 77 90 107] such as fear of job loss, were also cited as common barriers. Other less frequently mentioned barriers included protection of self (7/161),[24 36 37 81 91 127] avoidance of discussion in meetings (4/161),[8 67 86 108] and apprehension of sending an inappropriate form (1/161).[69]

### Process and systems of reporting

Process and systems of reporting was mentioned as a barrier to reporting 110 times. The most frequently identified barrier to incident reporting was the time required to complete an incident report (29/110),[8 11 27 38 39 41 52 56 62 67 68 71 72 76 77 79–81 84 87 95 108 109 111 116 119 120] followed by the complexity of the reporting process (28/110).[8 9 11 30 43 45 47 53 54 71 72 77 80 81 86 87 89 95 105 109–111 116 119 120] Other process and systems of reporting barriers included lack of anonymity and/or confidentiality in reporting (22/110),[8 11 24 27 36 37 47 56 58 68 70 71 80 81 91 102 105 108 120] reporting format (10/110),[39 43 53 73 77 86 111 116] and the type of reporting system (eg, paper-based) (5/110).[58 76 86 95] Less frequently

mentioned barriers included lack of information to complete report (3/110),[78 81 84] the focus of reporting (1/110),[71] and information to complete report not readily being available (1/110).[43]

### Incident characteristics

Incident characteristics were mentioned as a barrier to reporting 92 times. Level of harm, cause of incident and frequency of incident were the most frequent incident characteristics acting as barriers to reporting (40/92, 19/42 and 18/92, respectively). HCPs were less likely to report an incident if the patient experienced no or minimal harm.[8 11 24 30 31 33 35 37 39 40 43 47 51–56 58 59 63 66 67 76 84 90 92 93 101 103 105 108 109 116 119 120 122] Incidents that were deemed to occur frequently were considered too well-known to report.[30 36 40 41 43 66 69 74 80 84 88 91–93 103 116] Furthermore, if the cause of the incident was deemed unpreventable this acted as a barrier to incident reporting.[32 38–40 47 66 73 80 81 84 88 92 93 116 129] Other barriers included the type of incident (13/92),[8 32 38 39 41 45 46 67 76 77 81 86 116] and the level of risk (2/110).[11 63]

### Individual HCP characteristics

Barriers reflective of individual HCP characteristics were cited 89 times. Barriers included a negative attitude/lack of value placed on incident reporting (53/89),[8 9 36 38 40 41 47 53 54 61 65 66 68 72 76 77 79–81 86 87 92 98 100 102 103 105 107–109 116 119 122 126] and the perception that incident reporting does not result in improvements typically underlined such negative attitudes and values. A number of studies found that HCPs fail to report incidents because they simply forget (9/89),[8 27 35 43 77 86 88 93 108] and that the way HCPs perceive themselves can act as a barrier to reporting (9/89).[24 37 60 81 91 94 108] Less frequently mentioned barriers included emotional responses to the incident (6/89),[43 63 72 73 116] previous reporting behaviour (5/89),[32 46 48 68 97] exposure to errors (2/89),[95 114] and length of time in employment (2/89).[48]

### Knowledge and skills

Knowledge and skills were cited as barriers to incident reporting 84 times. The two most frequently mentioned barriers related to a lack of reporting clarity (36/84),[9 11 24 27 30 32 36 37 40 41 43 47 53 54 72 80 81 84 88 91 92 95 103 105 108 109 116 119] and a lack of clarity regarding what constitutes an adverse event and/or near miss (31/84).[9 11 30 39 41 43 47 52–54 67 68 73 76 77 79 86 108 109 112 116 119] This suggests that a lack of knowledge about what should be reported and how to do this act as barriers. Less frequently cited barriers included an inability in error recognition (7/84),[47 69 72 76 79 120 129] lack of training in reporting (5/84),[36 73 102 107 114] and lack of awareness (4/84).[47 52 84 120]

### Work environment

Work environment was mentioned 80 times as a barrier to incident reporting. Workload/priority (50/80),[9 11 24 27 30 35–37 40 43 46 47 52 56 57 60–63 67 69 70 73 76 77 86 88 89 91–93 98 102 103 106 109–111 116 126] and accessibility (27/80),[24 27 30 32 37 41 43 46 47 61 68 69 73 77 80 81 84 86 88 91 107 119 120] were the most frequently

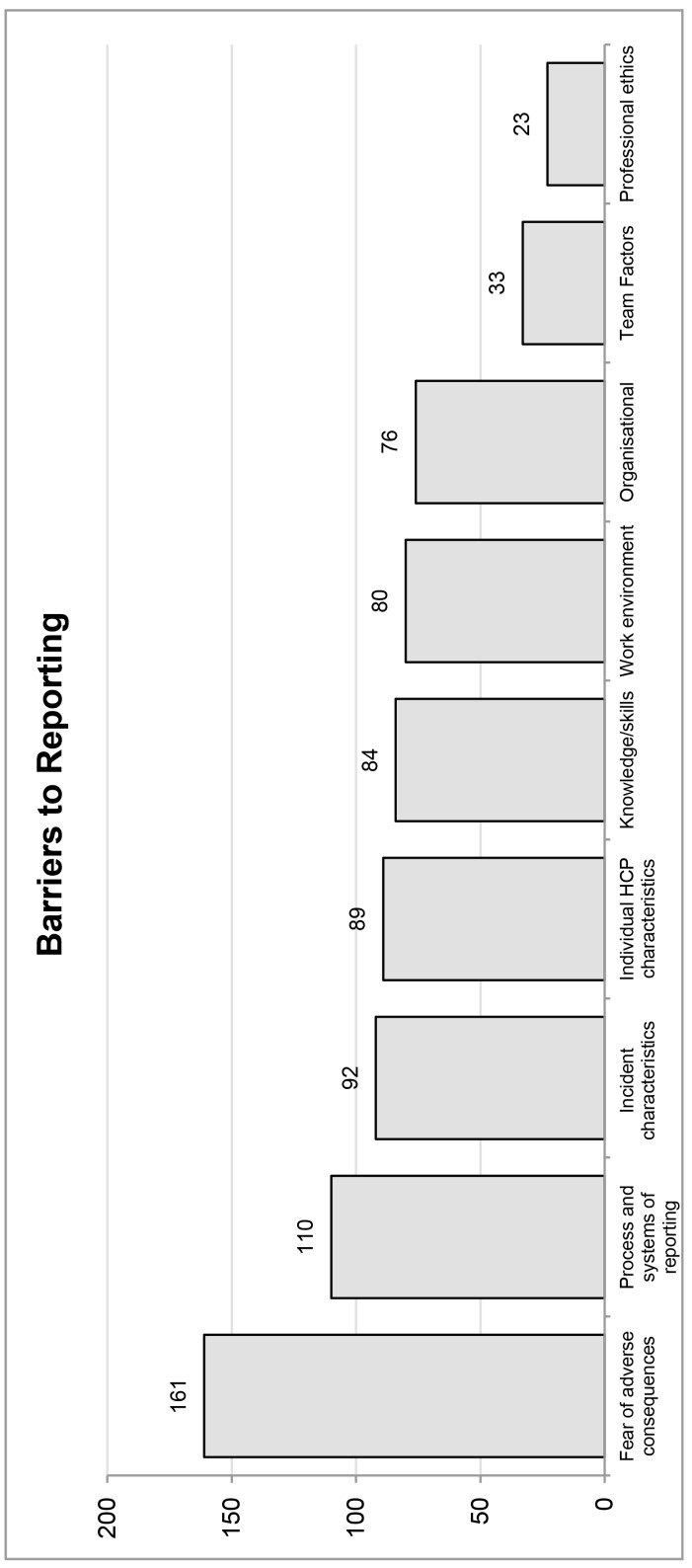

**Barriers to Reporting**

**Figure 3** Continued

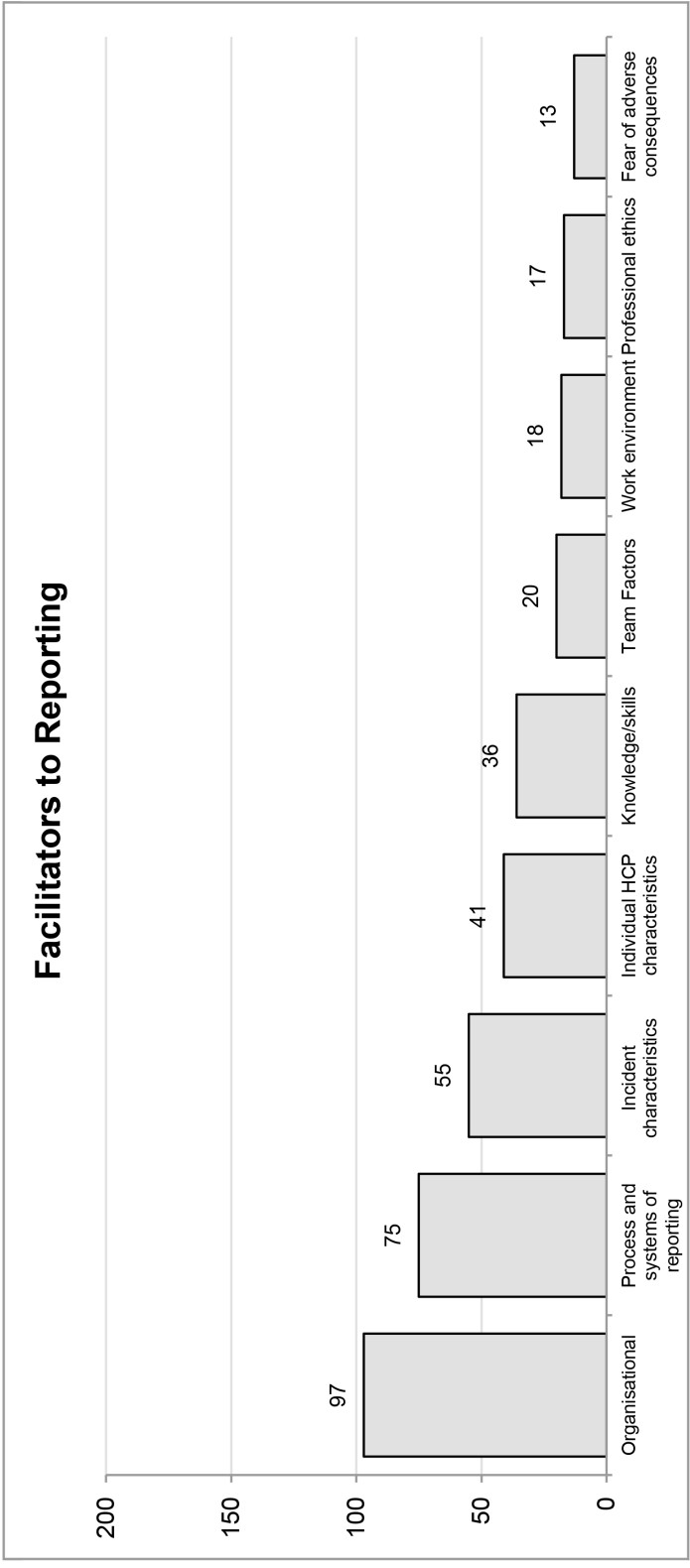

**Figure 3** Continued

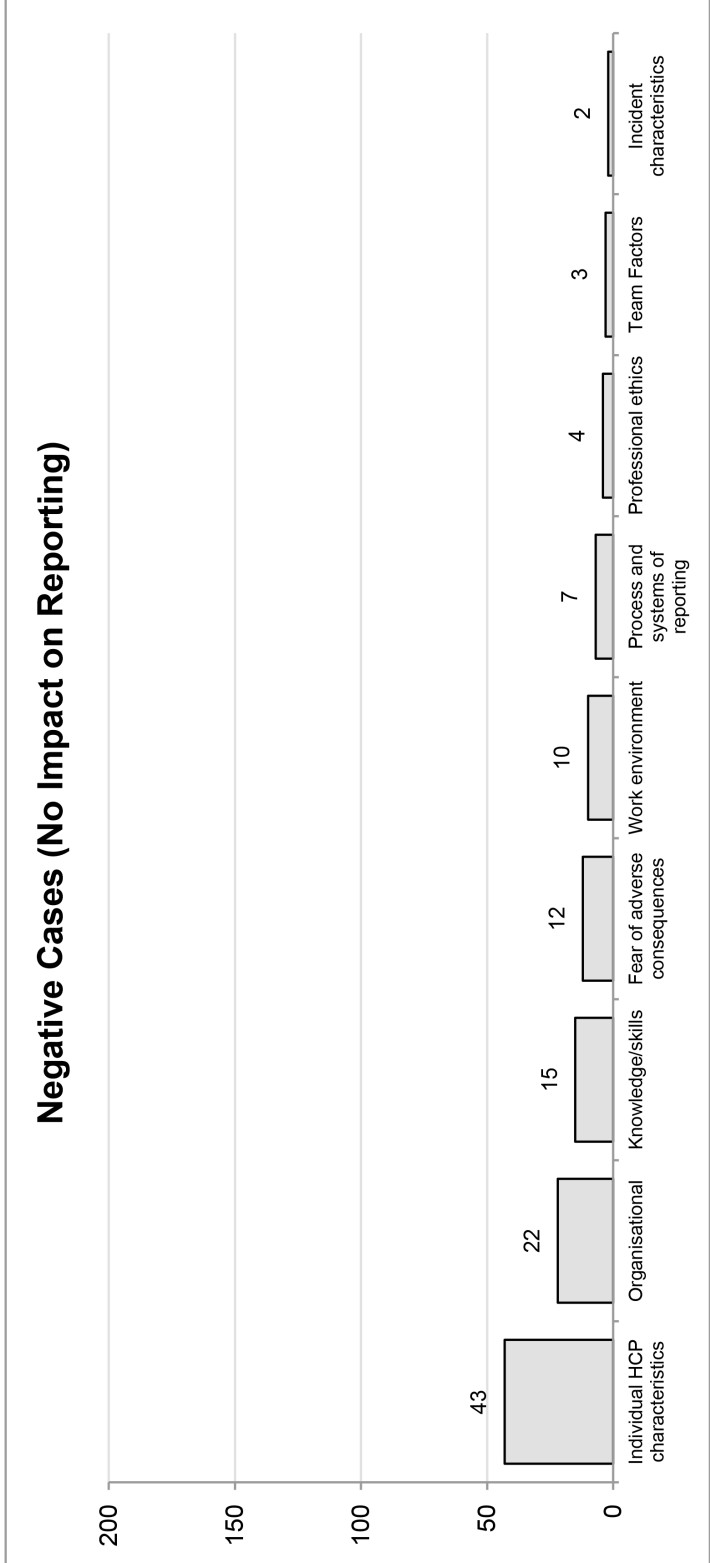

**Figure 3** Frequency of categories influencing engagement in patient safety incident reporting. HCP, healthcare professional.

mentioned work environment barriers, suggesting that high workload does not allow for incident reporting to be prioritised, and that access to the reporting system is problematic (eg, not enough computer work stations to access reporting forms).

## Organisational factors

Organisational factors were mentioned 76 times as a barrier to incident reporting. Lack of feedback and communication following incident reporting (26/76),[8 9 11 39 47 48 52 53 61 63 64 67 71 76 79 86 98 99 107 108 111 116 120 121 128] and the absence/lack of a positive reporting culture (17/76),[9 10 38 46 47 57 66 76 84 86 87 103 107 111 128] were the two most frequently mentioned organisational barriers to reporting. Less frequently mentioned were lack of organisational learning and improvement (7/76),[27 39 47 67 98 102 116] poor organisational use of data (7/76),[52 64 76 79 98] and poor management response to reports (5/76).[60 72 76 83 102]

## Team factors

Team factors were mentioned as barriers to engagement in incident reporting 33 times. The three most frequently mentioned barriers included the negative impact that incident reporting could have on working relationships (13/33),[11 27 44 60 63 66 68 108 109 111 116] the influence of seniors not to report (7/33),[48 51 68 73 82 120] and how HCPs feel about reporting their peers (5/33).[39 40 72]

## Professional ethics

Professional ethics was the least frequently mentioned barrier to incident reporting (23/748). The most prevalent factor was a lack of personal responsibility to report (15/23),[8 9 32 41 46 47 53 77 78 87 92 103 116 118] with studies suggesting that HCPs are less likely to report when they feel that reporting is the responsibility of someone else within the team. Concealment was also mentioned as a barrier (5/23).[39 108 126]

## Frequency of facilitators in patient safety incident reporting

Facilitators of reporting were mentioned 372 times across the 110 articles (see table 2). Organisational factors were the most frequently mentioned facilitator to incident reporting (97/372), followed by process and systems of reporting (75/372) and incident characteristics (55/372) (online supplementary table 1).

## Organisational factors

Organisational factors were mentioned as facilitators 97 times. The two most frequently cited facilitators included the provision of feedback/communication following incident reporting (29/97),[9 11 36 42 45 50 53 54 69 70 80 81 83 86 98 101–103 108 116] and a non-punitive incident reporting policy (22/97).[9 11 29 36 38 42 44 45 49 54 63 69 70 80 81 102 108 120] The existence of a reporting culture (16/97),[29 41 45 66 69 82 83 96 113 116 120 123 127] and a focus on learning and improvement from incidents (13/97),[9 39 43 49 82 98 102 103 111 116] were also facilitators to reporting.

## Process and systems of reporting

Process and systems of reporting was mentioned as a facilitator 75 times. Reporting format, ensuring anonymity and/or confidentiality, and simplification of reporting were the three most frequently cited facilitators accounting for 21/75,[9 11 25 42 53 54 63 69 81 86 98 101–103 108 116 120] 16/75,[9 11 29 43 49 53 68 86 101 102 108 116 120] and 15/75,[9 11 38 42 70 80 86 95 101 102 105 116] facilitators within this category. Less frequently mentioned process and systems of reporting facilitators included the type of reporting system used (eg, electronic reporting) (11/75).[45 46 49 53 80 86 102 105]

## Incident characteristics

Incident characteristics were mentioned as a facilitator to reporting 55 times. Level of harm and frequency of an incident were the most frequently cited incident characteristics identified as facilitators to reporting (26/55,[11 39 41 43 49 51 55 58 63 66 69 70 73 84 89 92 109 112 129] and 13/55,[11 41 66 69 70 84 129] respectively). Incidents resulting in severe harm (including death) were more likely to be reported and HCPs were more likely to report incidents that occur infrequently rather than frequently. Less frequently mentioned facilitators included the type of incident (8/55),[39 41 73] cause of the incident (6/55),[36 49 66 70 89] and level of risk (1/55).[63]

## Individual HCP characteristics

Individual HCP characteristics were mentioned 41 times as a facilitator. A positive attitude towards incident reporting and a high value placed on incident reporting was found to increase the likelihood of reporting (21/41).[9 11 49 63 73 77 81 89 102 109 111 112 114 115 123] HCPs' emotional response to a patient safety incident was also found to increase the likelihood of reporting in a number of studies (5/41).[43 63 116] The professional group of HCPs was also found to act as a facilitator to reporting (5/41).[28 104] Less frequently cited facilitators included previous reporting behaviour (1/41),[29] number of hours worked (1/41),[32] and demographics (eg, gender and age) (2/41).[48 115]

## Knowledge and skills

Training in reporting was identified as the most frequently mentioned facilitator in this category (21/36).[9 25 36 45 69 80 86 91 103 105 108 120] Other facilitators included knowledge regarding what constitutes an adverse event/near miss and the ability to recognise an error has occurred (7/36,[9 42 53 54 103 108 116] and 4/36,[36 69 70 129] respectively).

## Team factors

Team factors were mentioned 20 times as a facilitator to reporting. Good teamwork/communication (7/20),[69 70 96 127] and a positive team culture (4/20),[81 115 123 127] were the most frequently cited facilitators.

## Professional ethics

Professional ethics was cited as a facilitator 17 times. A strong sense of duty (8/17),[39 69 80 81 109 112] and responsibility (5/17),[70 75 78 111] to report increased the likelihood

of reporting. Less frequently cited facilitators included accountability (2/17),[41 109] and a legal obligation to report (1/17).[48]

## Work environment

Work environment was mentioned as a facilitator 18 times. Access to the incident reporting system (11/18),[42 68 69 80 86 102 105 108 116] and those whose workloads allowed for and those that prioritised incident reporting increased the likelihood of reporting.

## Fear of adverse consequences

Fear of adverse consequences was mentioned as a facilitator to reporting 13 times and included a fear of litigation and fear of blame increasing the likelihood of reporting (8/13,[9 11 27 45 73 109 111] and 4/13,[9 11 108 109] respectively).

## Frequency of negative cases

Negative cases were identified 118 times across the 110 articles (see table 2). The three most frequently mentioned factors included individual HCP characteristics (43/118), organisational factors (22/118), and knowledge and skills (15/118), (online supplementary table 1).

Individual HCP characteristics were mentioned as a negative case 43 times. HCPs' attitude and value of incident reporting did not have an impact on reporting behaviour (12/43).[33 35 48 56 72 93 113] Similarly, HCPs' demographics (eg, age, gender) had no impact on the likelihood of reporting (12/43).[30 32 48 57 70 89 93 113 114] Other less frequently mentioned factors included seniority (4/43),[48 70 89 93] forgetfulness (1/43),[93] previous reporting behaviour (1/43),[93] and number of hours worked (1/43).[26] Organisational factors were cited as having no impact on incident reporting 22 times. The most frequently mentioned were the ownership of the organisation (eg, private/public funded) (6/22),[25 70] and management response towards incident reporting (4/22).[29 114 125] Knowledge and skills were mentioned 15 times. These included the clarity of the reporting mechanism (5/15),[29 35 56 93] knowledge of what constitutes an adverse event/near miss (2/15),[48, 72] ability in error recognition (1/15),[56] and training in error reporting (7/15).[25 70 93 107]

Fear of adverse consequences was cited as having no impact on engagement in incident reporting 12 times. These included a fear of litigation (4/12),[24 49 56 111] a general fear of adverse consequences (3/12),[35 39 113] blame (1/12),[56] judgement (1/12),[80] and impact on career (1/12).[89] Work environment was mentioned as having no impact on reporting 10 times, including workload/priority (3/10),[30 89 128] and unit type (3/10).[57 83] Other less frequently cited work environment factors included physical work conditions (1/10),[26] satisfaction with work environment (1/10),[124] and accessibility (1/10).[56]

Across all studies, process and systems of reporting was mentioned seven times as having no impact on incident reporting; these included reporting format (3/7),[25 89 102] complexity/simplification of reporting (1/7),[102] and anonymity and/or confidentiality (1/7).[24] Professional ethics were only mentioned four times as having no impact on the likelihood of incident reporting; these were legal obligation (2/4),[48] duty (1/4),[89] and responsibility (1/4).[26] Team factors were cited as having no impact on the likelihood of reporting three times, including teamwork and communication (2/3),[128] and support/encouragement to report (1/3).[122] Incident characteristics were the least frequently mentioned factor which had no impact on reporting. Cause of incident was found to have no impact on engagement in reporting (2/2).[89 93]

## DISCUSSION

It has been suggested that there is a tendency in healthcare to encourage reporting of any and all patient safety incidents, to celebrate large quantities of incident reports and to aim for ever-increasing overall reporting rates. While there are numerous problems associated with this approach[7] (eg, flooding the system to such a degree that the thorough investigation of each incident reporting is unachievable), it is clear that high levels of under-reporting seriously compromise the ability of incident reporting systems to facilitate learning and improvements in patient safety.

This is the first theoretical literature review of factors contributing to patient safety incident reporting. Based on the evidence from 110 articles, we developed a theoretical framework, based on the principles of grounded theory, which summarises a wide range of factors contributing to incident reporting. We purposely sought publications from a range of countries, covering diverse health systems and study populations with a view to incorporating these into one broad theoretical framework. We argue that this is an appropriate approach for this initial explorative work, as multiple theoretical frameworks for individual counties, settings and populations (eg, nurses working in mental health settings in Australia), would have limited application at this point in time. However, we suggest that those interested in exploring barriers and facilitators in specific settings conduct further research using the theoretical framework presented here.

To improve incident reporting (both the quantity and/or quality) and facilitate the successful implementation of incident reporting systems, we suggest that the theoretical framework is best used to prospectively and systematically identify factors within a given context that are likely to affect incident reporting. Those responsible for the effective implementation of incident reporting systems should explore each of the factors listed in our framework for salience. Rather than the framework being used in isolation, we recommend that it be used in conjunction with other implementation theories/frameworks and models to guide, understand and evaluate implementation of incident reporting systems.[130] Based on such prospective analysis, strategies to enhance the adoption, implementation and sustainability of incident reporting systems can

be tailored and selected according to a given setting. As such, using the developed framework will advance our understanding of how to optimally implement incident reporting systems into practice.

We used the developed theoretical framework, based on the evidence base, to organise our findings and have presented the frequency and rank order (ie, prevalence) of factors contributing to incident reporting. While this approach is consistent with other frameworks in the patient safety literature,[14 23] it may be considered as a crude analysis of the existing literature and needs to be interpreted with caution. We acknowledge that it is possible, although unlikely, that a relationship between the number of times a given factor is mentioned in the literature and its impact on incident reporting behaviour might not exist. However, we have been able to provide the first high-level overview of a large heterogeneous body of evidence. Furthermore, we acknowledge that weighting the impact of each factor would have been advantageous; however, the data did not lend itself to this possibility and we propose that it might not be possible to simply weight factors because of the complex and dynamic interrelationships that are likely to exist between them. Alternatively, we suggest that modelling the interrelationships between factors affecting incident reporting engagement is an avenue for future research.

Our results suggest that fear of adverse consequences and ineffective processes/systems of reporting are high-priority areas that require consideration to improve engagement in incident reporting. Changes to policy should be considered at an institutional or national level to prevent fear of litigation and blame, as fear of adverse consequences was found to inhibit incident reporting. We believe that it is unlikely that changes made within a single hospital or healthcare system would instil significant reassurance to promote incident reporting. In addition, at an organisational level we found that appropriate systems and processes for reporting need to be implemented to improve incident reporting; simultaneously, lack of or poorly designed systems significantly hinder reporting. These aspects of reporting rely on well-designed processes and technologies and are arguably the responsibility of organisational leaders. There is no 'optimum model' for incident reporting systems (eg, electronic, confidential, anonymous)—systems need to be responsive to users and organisational needs.

Organisational factors and processes/systems of reporting were identified as the two most frequently cited facilitators of reporting, which suggests that healthcare organisations consider these as high-priority areas which should be the target of increased focus and resources. For example, our results suggest that organisational policies that foster a reporting and learning culture as well as providing feedback following a report will promote incident reporting. Interestingly, we found that individual HCP characteristics have little impact on engagement in incident reporting. This suggests that organisations should be cautious before investing significant resources in these factors as such investment may result in minimal returns.

Although we have considered the above factors in isolation as illustrative examples, it is important to consider the interconnecting relationships between factors in order to develop intervention packages to improve engagement in incident reporting. Our results suggest that a comprehensive intervention/policy package which targets more than one contributing factor (eg, establishing a supportive work environment, with mechanisms which optimise shared learning, alongside a national policy to minimise the fear of adverse consequence) is far more likely to result in increased engagement in incident reporting compared with interventions that simply target one factor.

## Strengths and limitations

In order to identify as much relevant literature as possible, we have included quantitative, qualitative and mixed-methods research and have not restricted the literature to specific incident reporting systems, that is, departmental, local, regional and national. In addition, the studies included a vast array of healthcare settings and providers, maximising the generalisability of the results. The resulting evidence has been synthesised into a practical output, that is, a theoretical framework to guide efforts to improve engagement in incident reporting.

The results and recommendations proposed in this evidence synthesis must be considered in light of several limitations. First, only articles published in English were included, which may generate bias. However, articles spanning 4 continents from >20 countries were identified, hence we are confident that our findings are of high external validity to guide safety policy globally. Second, the last systematic search for literature was conducted on 29 May 2014, meaning that literature published since this date will not have been included. We suggest that literature published after the last search could be useful to test the validity of the theoretical framework. Third, the decision not to include studies detailing interventions to improve incident reporting and studies detailing variations in engagement in incident reporting may skew the findings. This decision was made as it was not possible to determine the relative contribution of individual factors on engagement in incident reporting within such studies. Fourth, large heterogeneity across studies in terms of outcome measures and methodologies meant conduction of meta-analysis was precluded. This having been said, the synthesis of barriers and facilitators into frequency of reporting provides some evidence towards their respective relative importance, although it is accepted that the frequency of factors may represent those that have been the subject of more research. We recommend that future research applies and evaluates the usefulness of the developed theoretical framework in exploring and improving incident reporting in a variety of settings (eg, primary and secondary healthcare).

## Future research

There are many ways in which future research could test the validity of the theoretical framework presented in the current study. For example, content validity of the theoretical framework could be assessed using expert consensus methods (eg, Delphi study). In addition, predictive validity could be tested quantitatively by assessing the correlation between, for example, fear of adverse consequences (level of fear) and incident reporting behaviour (ie, number of incidents reported). A negative correlation between number of incidents reported (low) and fear of adverse consequence (high) would provide evidence for predictive validity of the theoretical framework.

## SUMMARY/CONCLUSION

A wide range of factors contributing to engagement in incident reporting exist across varying levels of the healthcare system. Efforts aimed at addressing the current tendency to under-report must consider the full range of factors in order to develop tailored interventions and policy packages for improvement. We suggest the theoretical framework developed here would be useful in understanding factors affecting incident reporting engagement, increasing engagement in incident reporting and ultimately learning from patient safety incidents.

**Acknowledgements** The authors thank Kelsey Flott, BA, MSc, from the Centre for Health Policy, Imperial College London, for critically reviewing the manuscript.

**Contributors** Conception or design of the work: SA, LH, EM, NS and AD. Data collection: SA, LH and TS. Data analysis and interpretation: SA, LH, TA and NS. Drafting the article: SA, TA and NS. Critical revision of the article and final approval of the version to be published: SA, LH, TS, EM, TA, NS and AD.

**Funding** This article represents independent research supported by the National Institute for Health Research (NIHR) Imperial Patient Safety Translational Research Centre. LH and NS' research was supported by the National Institute for Health Research (NIHR) Collaboration for Leadership in Applied Health Research and Care (CLAHRC) South London at King's College Hospital NHS Foundation Trust. LH and NS are members of King's Improvement Science, which is part of the NIHR CLAHRC South London and comprises a specialist team of improvement scientists and senior researchers based at King's College London. Its work is funded by King's Health Partners (Guy's and St Thomas' NHS Foundation Trust, King's College Hospital NHS Foundation Trust, King's College London and South London and Maudsley NHS Foundation Trust), Guy's and St Thomas' Charity, the Maudsley Charity and the Health Foundation.

**Disclaimer** The views expressed are those of the author(s) and not necessarily those of the NHS, the NIHR or the Department of Health.

**Competing interests** None declared.

**Provenance and peer review** Not commissioned; externally peer reviewed.

**Data sharing statement** All data from this review and theoretical framework are presented within the publication.

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
