## [Reviewer comments · BMJ Open]

ARTICLE DETAILS

TITLE (PROVISIONAL)	Development of a Theoretical Framework of Factors Affecting Patient Safety Incident Reporting: A Theoretical Review of the Literature
AUTHORS	Archer, Stephanie; Hull, Louise; Soukup, Tayana; Mayer, Erik; Athanasiou, Thanos; Sevdalis, Nick; Darzi, Ara

VERSION 1 - REVIEW

REVIEWER	Patrice François Professor in Public Health University of Grenoble Alpes France
REVIEW RETURNED	03-May-2017

GENERAL COMMENTS	The article focuses on a topic of major importance in the field of patient safety management. This is a thorough review of the literature on the factors that influence the adverse events reporting by health care professionals. The method is well described and rigorous. We understand that a meta-analysis was not done due to the heterogeneity of studies' methods and outcomes. The major contribution of the study is to develop a theoretical framework to classify factors. This framework seems relevant and particularly useful for future studies on the same issue. A detail: page 10 line 38, a sentence appears incomplete.
---

REVIEWER	Julie Polisena CADTH, Canada
REVIEW RETURNED	03-Jun-2017

GENERAL COMMENTS	This paper is well-written, and the topic is germane. 1) My main critique of this paper relates to the literature search timeframe. I suggest that the authors conduct a search update to identify potentially relevant systematic reviews and primary studies published since 2014. 2) In the Methods section, I suggest that the authors briefly describe the theoretical review approach. 3) Line 39 on page 10, the sentence, "We do not...", is incomplete.
--

	4) In the Discussion, the readers may be interested to know the general steps to assess the validity of the theoretical framework in future research. A brief description would be interesting.
--	---

VERSION 1 – AUTHOR RESPONSE

Reviewer: 1

The article focuses on a topic of major importance in the field of patient safety management. This is a thorough review of the literature on the factors that influence the adverse events reporting by health care professionals. The method is well described and rigorous. We understand that a meta-analysis was not done due to the heterogeneity of studies¹ methods and outcomes. The major contribution of the study is to develop a theoretical framework to classify factors. This framework seems relevant and particularly useful for future studies on the same issue.

Many thanks for the positive comments on our work.

A detail: page 10 line 38, a sentence appears incomplete.

This was a typographical error that we have now corrected.

Reviewer: 2

This paper is well-written, and the topic is germane.

1) My main critique of this paper relates to the literature search timeframe. I suggest that the authors conduct a search update to identify potentially relevant systematic reviews and primary studies published since 2014.

We appreciate that the last search was conducted in May 2014 and acknowledge that some recent papers may have met the inclusion criteria. Whilst updating the search would be ideal and would likely identify more articles, we are currently unable to update the search due to logistical constraints – effectively the team that worked on this review has disbanded and researchers have moved onto other research projects. Although this is a major logistical problem, we remain confident in the thematic results and overarching conclusions that we report:

- Considering the significant number of articles (>100) spanning over two decades (24 years) that are included in the current review, we are confident that the conclusions drawn and principles of the theoretical framework are valid.
- Updating the search would generate further studies – but due to the inclusiveness of the review the constructs contained in the theoretical framework would likely remain.

We do appreciate that this is a limitation of the review. We thus suggest that studies published after the last search could be useful in testing the validity of the theoretical framework – for example, Improving Incident Reporting Among Physician Trainees by Krouss et al (2016) published in the Journal of Patient Safety.

As such, we have included the following sentence: 'We suggest that literature published after the last search could be useful to test the validity of the theoretical framework'. (p.27) Lastly, we have highlighted this as an explicit limitation in the discussion to ensure readers are aware of when the last search was performed and the fact that additional articles, published after the last search are likely to exist: 'The last systematic search for literature was conducted on 29/05/2014, meaning that literature published since this date will not have been included'. (p.27).

2) In the Methods section, I suggest that the authors briefly describe the theoretical review approach.

We thank the reviewer for this comment. We agree that it was important to describe the approach taken to the development of the theoretical framework, and that this should be to a level that complements the rest of the manuscript. The following section was thus developed and included within the manuscript to provide appropriate detail for the reader.

A grounded theory approach was used to guide the development of the theoretical framework. Grounded theory is associated with the discovery of theory from data systematically obtained from social research.[21] It has been identified as a method where thorough and theoretically relevant analysis of a topic can be reached, specifically within literature reviews.[22] In light of this, a three-stage approach was undertaken to develop a theory of factors contributing to engagement in patient safety incident reporting. The first stage, coding, includes identifying parts of the data that relate the phenomena in question (in this case, incident reporting). During this stage, known as open coding in the grounded theory literature, three authors (SA, LH & TS) read and re-read each paper and identified sections of the paper that were relevant to the research question. Initial concepts developed from these were noted down at this stage; in some cases these were consistent with pre-existing literature (e.g. in the case of a standardised scale), but in others allowed for unseen insights to develop across the data corpus (e.g. in qualitative studies). In the second stage, conceptualising, or axial coding, focused on grouping together the initial codes where there were relationships to form higher order categories. These were given names. Stage three, categorising, or selective coding focused on linking together similar higher order categories that contained similar concepts which could underpin the reasoning behind the way that the phenomena (in this case, incident reporting) could be explained. Figure 1 displays an example of how these stages were applied.

Engagement in these three stages allowed constant comparison between the articles in the dataset to be performed until a theoretical framework was confirmed. The final theoretical framework was reviewed by another member of the research team (NS) and feedback regarding the category descriptors was incorporated. The final theoretical framework of factors contributing to patient safety incident reporting engagement is displayed in Table 2.

The theoretical framework developed was used to organise the identification of factors found to affect incident reporting and to quantify their prevalence. This approach is consistent with existing frameworks in the patient safety literature, for example Lawton et al employed a similar approach to quantify the prevalence of factors contributing to patient safety incidents in hospital settings.[23]

3) Line 39 on page 10, the sentence, "We do not...," is incomplete.

This was a typographical error that we have now corrected.

4) In the Discussion, the readers may be interested to know the general steps to assess the validity of the theoretical framework in future research. A brief description would be interesting.

Thank you for this useful suggestion. We agree that readers would be interested in knowing how they could assess the validity of our theoretical framework and we have expanded on this aspect with the section below.

There are many ways in which future research could test the validity of the theoretical framework presented in the current study. For example, content validity of the theoretical framework could be

assessed using expert consensus methods (e.g. Delphi study). In addition, predictive validity could be tested quantitatively by assessing the correlation between, for example, fear of adverse consequences (level of fear) and incident reporting behaviour (i.e. number of incidents reported). A negative correlation between number of incidents reported (low) and fear of adverse consequence (high) would provide evidence for predictive validity of the theoretical framework.